# Performance Analysis of Relay-Aided NOMA Optical Wireless Communication System in Underwater Turbulence Environment

Yanjun Liang , Hongxi Yin *, Lianyou Jing, Xiuyang Ji and Jianying Wang

School of Information and Communication Engineering, Dalian University of Technology, Dalian 116024, China
* Correspondence: hxyin@dlut.edu.cn

**Abstract:** Non-orthogonal multiple access (NOMA) is a promising technology to improve spectrum utilization effectively for underwater optical wireless communications (UOWC). To exploit the benefits of NOMA in a turbulent environment, cooperative transmission has been introduced in the NOMA–UOWC network. The existing studies on NOMA suggest that relay selection and power optimization are the main factors affecting system performance. In this paper, a general NOMA node pairing method and two power optimization schemes for NOMA–UOWC are proposed, and both schemes are proven to be strictly quasi-convex. The two optimization schemes are solved by the BFGS algorithm and the particle swarm algorithm, respectively. The effectiveness of the proposed schemes are evaluated by our simulations, and the main factors affecting the relay-aided NOMA performance are derived.

**Keywords:** non-orthogonal multiple access; underwater optical wireless communications; turbulence channel; relay selection; power allocation optimization

## 1. Introduction

Underwater optical wireless communication (UOWC) will be employed as a vital component of the marine internet of things (IoT) in the future due to its high bit rate, ultra-low latency, and high security [1]. Many researchers have carried out work on UOWC, and various theoretical and experimental studies have been conducted to investigate the behavior of optical beams underwater [2–4]. In the meantime, UOWC still confronts many challenges; e.g., the complex noise sources in the ocean environment and the severe channel fading caused by absorption, scattering, turbulence, and bubbles in the seawater [3,4]. On the other hand, most current underwater communication devices use light-emitting diodes (LEDs) as light sources or transmitters, for cost reasons, and the limited modulation bandwidth of LED limits the performance improvement of UOWC systems to some extent [5]. Non-orthogonal multiple access (NOMA) technology could be an attractive alternative to overcome these drawbacks.

There have been notable researches that demonstrated the ability to improve system spectrum utilization of NOMA in radio frequency (RF) networks and free-space optical networks [6,7]. As for underwater networks, most of the research on NOMA focuses on optimizing the receiving rate and improving the communication performance of the system through reasonable resource management, while researches on the exact underwater communication channel are investigated infrequently [8–10]. In [8], a practical NOMA scheme for a visible light communication (VLC) system on land was proposed, which improved spectral efficiency by 50% compared to orthogonal frequency division multiple access (OFDMA). An underwater optical wireless cellular network based upon NOMA was presented in [9], and the closed-form expressions of ergodic capacity for the near-end and the far-end receiving nodes in a single NOMA group were deduced. The influence of weak ocean turbulence on the performance of the underwater optical NOMA system was

studied in [10], and an optimizing problem joining the user–subcarrier pairing with power allocation was formulated to maximize the system utility. Nevertheless, the fairness of the far-end receiving nodes in the NOMA system was neglected. In addition, the effectiveness and reliability of a terrestrial NOMA communication system can be appropriately balanced through flexible power control and interference suppression; however, there still exist many issues to be explored before this technology can be employed in the underwater wireless optical NOMA communication network in the complex underwater noise and channel environment.

Nevertheless, the superimposed transmission of signals has inevitably contributed to the increase of inter-user interference, which imposes restrictions on the performance of the NOMA communication system [11]. Motivated by the fact that the cooperative communication technology can improve transmission reliability of communication networks markedly by means of relay forwarding [12], the relay-aided NOMA scheme may be more consistent with practical applications. Nevertheless, the research on power allocation algorithms is mostly focused on the improvement of the FPA (fixed power allocation) method based on the characteristics of the channel to date, which distinctly lack flexibility in optimizing the performance of the NOMA–UOWC system.

Simultaneously, many studies in recent years have been implemented for relay-assisted UOWC. Most of the current research has been based on the scattering and absorption characteristics of the underwater channel [13–16], with hardly any consideration of background ocean noise [17]. Ref. [13] discusses the requirements and design challenges for relay-assisted UOWC. In [14], the attenuation and fading characteristics of UOWC in turbulent channels were systematically analyzed, and the performances of amplified forwarding (AF) and decoded forward (DF) multi-hop relay networks were investigated, respectively. The connectivity of the UOWC network based on the multi-hop transmission was investigated in [15,16]. An optimization strategy for relay deployment in the UOWC network was proposed in [17], and a resource allocation optimization algorithm was designed to uniformly manage the power allocation for relay nodes.

For the underwater optical network, there have been few researches on the combination of NOMA and cooperative communication technology. In [18], a cooperative NOMA system was proposed with the near-end node playing a role of relay node. However, this study ignores the direct transmission path between the source node and the receiver node 2, and it does not discuss the combined reception of this part of the optical signal. The above researches did not consider the combination of NOMA and the independent relay nodes, nor did they take into account the more realistic and complex underwater channel environment, such as solar radiation background noise, etc.

In this paper, the relay selection and power allocation technologies of a relay-aided NOMA–UOWC network are deeply explored, focusing on the combination of NOMA and cooperative communication technologies. In consideration of the influence of underwater solar radiation noise and turbulent channels, the cooperative transmission model of UOWC based on NOMA is established, and a user pairing strategy for joint relay selection is proposed. A global optimal power allocation algorithm (GOPA) and a stepwise sub-optimization power allocation algorithm (SSOPA) are designed for the relay-assisted NOMA communication network based on AF. The effectiveness of the proposed strategy and algorithm is verified by our simulations.

## 2. System Model

The deployment of nodes is constrained by the communication range in the seawater environment, and an overly long distance between communication nodes will lead to unstable optical communication links. In this paper, we consider the cooperative transmission of UOWC based on AF relay, since such a relaying technique is comparatively easy to implement and its computational complexity is relatively low. The diagram of the relay-aided NOMA–UOWC system is depicted in Figure 1. An underwater optical network

is taken into account, in which a transmitter serves $N$ receiving nodes simultaneously with the assistance of $M$ relaying nodes.

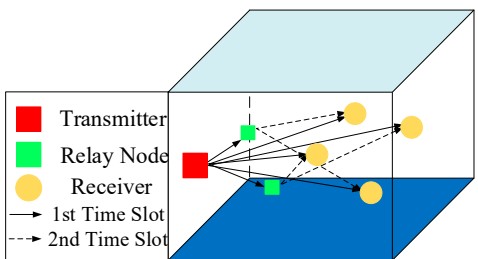

**Figure 1.** Architecture of the relay-aided NOMA–UOWC system.

According to the principle of NOMA, the user pairing and the relay selection need to be accomplished in real-time before the transmission [19]. Firstly, the channel gains [20,21] between the communication nodes can be calculated by:

$$
h = \underbrace{\alpha^2}_{\text{attenuation}} \underbrace{\exp(-cd)}_{\text{transmission loss}} \underbrace{\frac{A_{\text{de}}\eta_{\text{tr}}\eta_{\text{de}}\cos\theta}{2\pi d^2(1-\cos\theta_0)}}_{\text{geometric loss}},
\tag{1}
$$

where $c = \alpha + b$ is the attenuation coefficient of seawater, and $\alpha$ and $b$ are the absorption coefficient and the scattering coefficient of seawater, respectively. The optical properties for four typical seawaters are listed in Table 1 [1,22]. $d$ indicates the transmission distance of the optical signal, $A_{\text{de}}$ expresses the aperture area of the optical receiver, $\eta_{\text{tr}}$ and $\eta_{\text{de}}$ signify the transmitter optical efficiency and the receiver optical efficiency, respectively. $\theta$ is the tilt angle between the optical axis of the transmitter and the receiver, and $\theta_0$ represents the light divergence angle of the transmitter. $\alpha$ represents the attenuation of the optical signal caused by the turbulence. Here, the weak turbulent channel is taken into consideration, so the probability density function (PDF) of $\alpha$ can be expressed as [23,24]:

$$
f(\alpha) = \frac{1}{2\alpha\sqrt{(\pi\ln(\sigma_{\text{sci}}^2+1))/2}}\exp\left(-\frac{2(\ln\alpha-\mu_{\text{s}})^2}{\ln(\sigma_{\text{sci}}^2+1)}\right),
\tag{2}
$$

where $\sigma_{\text{sci}}^2$ indicates the scintillation coefficient, and $\ln\alpha$ obeys a Gaussian distribution with a mean value of $\mu_{\text{s}}$ and variance of $(1/4)\ln(\sigma_{\text{sci}}^2+1)$.

**Table 1.** Typical values of attenuation coefficient for seawater.

| Water Type | $\alpha$ (m$^{-1}$) | $b$ (m$^{-1}$) | $c$ (m$^{-1}$) |
|---|---|---|---|
| Pure seawater | 0.053 | 0.003 | 0.056 |
| Clean seawater | 0.114 | 0.037 | 0.151 |
| Coastal seawater | 0.179 | 0.219 | 0.398 |
| Harbor water | 0.295 | 1.875 | 2.170 |

Without loss of generality, the channel state information (CSI) between communication nodes satisfies $h_1 > h_2 > \ldots > h_n > \ldots > h_N$, and $h_n (n \in [1, N])$ indicates the optical channel gain from the transmitter to the $n$th receiving node. It is assumed that the $m$th relay node is chosen as the cooperative relay by the pairing NOMA receiving nodes $i$ and $j$, the superimposed signal transmitted by the transmitter to the relay node at the first time slot can be provided by [25]:

$$
x_{\text{t}}^m = \sqrt{a_i^m P_{\text{t}}^m}x_i + \sqrt{a_j^m P_{\text{t}}^m}x_j,
\tag{3}
$$

where $P_t^m$ represents the transmitted power when the $m$th relay node is chosen, $a_i^m$ and $a_j^m$ are the power allocation coefficients of the paired users, and $x_i$ and $x_j$ represent the modulated signals of the $i$th node and the $j$th node. The signal received by the $m$th relay node can be written as:

$$y_m = h_t^m \left( \sqrt{a_i^m P_t^m} x_i + \sqrt{a_j^m P_t^m} x_j \right) + \overline{n}_t^m, \tag{4}$$

where $h_t^m (m \in [1, M])$ is the channel gain between the transmitter and the $m$th relay node, and $\overline{n}_t^m$ signifies the noise between the transmitter and the $m$th relay node. Unlike terrestrial communications networks (TCN), there exist multiple noise sources in a marine environment. We mainly consider solar radiation noise, dark current noise, and shot noise as the main noises; therefore, the following expression can be obtained as [26,27]:

$$\overline{n} = \underbrace{2Bq\eta_{de}A_{de}\pi(\theta_{fov})^2 \Delta\lambda T_F L_{sol}}_{\text{solar noise}} + \underbrace{2Bq\eta_{de}P_{t_m}}_{\text{shot noise}} + \underbrace{4Bk_B T_e \varphi_F / R_{lr}}_{\text{thermal noise}}, \tag{5}$$

where $B$ signifies the system bandwidth, $q$ is the elementary charge, $\Delta\lambda$ indicates the optical filter bandwidth, $T_F$ represents the transmittance of the optical filter, $L_{sol}$ expresses the solar irradiance, $T_e$ is the equivalent temperature, $\varphi_F$ represents the system noise figure, $R_{lr}$ is the load resistance, $\theta_{fov}$ indicates the field of view for the receiver, and $k_B$ denotes the Boltzmann constant.

Utilizing AF mentioned above, the signal forwarded by the $m$th relay node can be expressed as:

$$x_m = \delta\eta_{tr}y_m, \tag{6}$$

where $\delta$ represents the amplification coefficient of the relay node. In terms of the principle of AF, the amplification coefficient, $\delta$, can be obtained as follows [28]:

$$\delta = \frac{P_m}{\eta_t \eta_{de}(h_m P_t^m + n_m)}, \tag{7}$$

where $P_m$ indicates the transmitted power of $m$th relaying.

As shown schematically in Figure 1, the receiving node receives the transmitted signals from the transmitter and the relay node in two-time slots under the line-of-sight (LOS) links. The maximal ratio combining (MRC) is employed at the receiver [29]. With synchronizing adopted, the received signal at the $i$th receiving node is:

$$y_i = \tau_1(h_i x_t^m + \overline{n}_i) + \tau_2(h_{m,i} x_m + \overline{n}_{m,i}), \tag{8}$$

where $\tau_1$, $\tau_2$ represent the combination factors, and $h_{m,i}$ indicates the channel gains between the $i$th receiving nodes and the $m$th relay node. The received signal at the $j$th receiving node can be calculated the same way. The stronger user restores the signal by performing success interference cancellation (SIC) to remove the signal of the weak user. The received signal-to-noise ratios obtained by the $i$th and the $j$th nodes can be respectively formulated as:

$$\Gamma_i^m = \frac{\left(a_i^m P_t^m h_i / \overline{n}_i\right)^2}{\left(1 + a_i^m P_t^m h_i / \overline{n}_i\right)^2} + a_i^2 \frac{\left(P_t^m h_m / \overline{n}_t^m\right)^2 \left(P_m h_{m,i} / \overline{n}_{m,i}\right)^2}{\left(a_j^m P_m h_{m,i} P_t^m h_m / \overline{n}_{m,i} \overline{n}_t^m + P_t^m h_m / \overline{n}_t^m + P_m h_{m,i} / \overline{n}_{m,i}\right)^2}, \tag{9}$$

$$\Gamma_j^m = \left(\frac{a_j^m P_t^m h_j}{\overline{n}_j}\right)^2 + \frac{\left(a_j^m P_t^m h_m / \overline{n}_t^m\right)^2 \left(P_m h_{m,j} / \overline{n}_{m,j}\right)^2}{\left(P_t^m h_m / \overline{n}_t^m + P_m h_{m,j} / \overline{n}_{m,j}\right)^2}. \tag{10}$$

## 3. User Pairing Scheme Jointed with Relay Selection

The first and most important issue in conducting the user pairing study is how many users access a NOMA group, which can implement optimal system performance. More

users accessing the NOMA group implies more users sharing power resources. The current mainstream studies are mostly based on two-user paired NOMA, the multi-user pairing is realized by adding new user accesses to the two-user paired group. Hence, the performance of multi-user paired NOMA is highly dependent on the channel quality of the newly added users, according to the NOMA principle. Taking a three-user paired NOMA system as an example, the system performance is improved only when the channel quality of the newly added users is simultaneously better than all users in the original NOMA group. In addition, as the number of pairing users increase, the probability of miscommunication increases geometrically. Therefore, this study focuses on the situation in which two users are paired considering the reliability of the system.

Although the user pairing technique can improve the performance of NOMA systems, the existing user pairing scheme in TCN is not completely applicable to the UOWC network with the combination of NOMA and cooperative communication techniques. In this section, an algorithm of user pairing joint relay selection is proposed, which can maximize the edge user communication rate while ensuring the central user communication requirement based on comprehensively considering the CSIs of the transmitter to receiving NOMA nodes and the relay nodes to receiving nodes.

In accordance with the analysis of the system model above, the transmitter selects an optimal relay node to forward the pairing NOMA signal according to the CSIs of the central user $i$ and the relay nodes. The selection of the relay node can be expressed by:

$$m = \arg \max_{r \in [1,M]} |h_{r,i} + h_i|. \tag{11}$$

This algorithm preferentially ensures the communication requirement of the NOMA central user. Based on the principle of NOMA, the set of edge nodes, $C_{ed}$, can be sorted by:

$$C_{ed} = \left\{ j^* : \ i < j^* \leq N, B \log_2\left(1 + \Gamma_{j*}^m\right) \geq R_{th} \right\}, \tag{12}$$

where $R_{th}$ indicates the threshold of the data rate. Accordingly, the NOMA pairing edge user $j$ is elected from $C_{ed}$ through (13):

$$j = \arg\max_{k \in C_{ed}} \left\{ \log_2(1 + \Gamma_k^m) + \log_2(1 + \Gamma_i^m) \right\}. \tag{13}$$

It is obvious that the pairing NOMA group in the proposed scheme can obtain the maximum sum rate while ensuring a certain quality of service.

## 4. Sum Rate-Based Power Optimization

The deployment of UOWC nodes is one of the most significant current discussions in the construction of marine IoT. Since the complexity of secondary replacement for batteries is very high, energy efficiency is an important consideration for the UOWC network where the nodes are powered with finite energy. In this section, two optimization schemes have been proposed to maximize the system sum rate with limited energy consumption while supporting a certain communication requirement.

### 4.1. Fixed Power Allocation Method

One of the most commonly used power allocation algorithms in a practical NOMA network is the FPA method [30,31]. The fixed power allocation algorithm's central mechanism is to assign a fixed power allocation ratio to all NOMA pairing users. Each user is allocated a proportionate share of the transmitted power.

In this case, we set the power allocation ratio as $\xi$. The optimization problem can be expressed as follows:

$$\max \ R_{total} \tag{14a}$$

$$\text{subject to } a_j^m = \xi a_i^m \tag{14b}$$

$$a_j^m + \xi a_i^m = 1 \tag{14c}$$

$$\sum_{m=1}^{M} \left( P_{\text{t}}^m + P_m \right) \leq P_{total} \tag{14d}$$

$$P_{\text{t}}^m > 0, P_m > 0 \tag{14e}$$

It is noted that the average allocation with the lowest algorithmic complexity has been employed for the bandwidth allocation between the NOMA groups. For the sake of providing a fair comparison, the results obtained by using different multiple access schemes are evaluated with the same bandwidth allocation method.

Hence, the total sum rates of the system in Equation (14c) can be calculated by the following equation:

$$R_{\text{total}} = \sum_{m=1}^{M} \frac{B}{M} \left( \log_2 \left( 1 + \Gamma_j^m \right) + \log_2 (1 + \Gamma_i^m) \right). \tag{15}$$

The algorithm is designed to be simple and easy to implement. The fairness of the NOMA system can be adjusted by adjusting the power allocation ratios. The specific algorithm implementation can be referred to in Algorithm 1. However, the algorithm ignores the NOMA group's users' channel conditions and cannot traverse all possible power allocation schemes. The step ratio configuration is critical to the optimization solution. The complexity of the system algorithm will increase if the step factor is too small, and the optimal solution of the algorithm cannot be found if the step factor is too large. While solving the optimization problem of (14d), the FPA algorithm, on the other hand, cannot solve the power distribution between the relay node and the source node directly. Hence, we employ the average distribution method to allocate the power between the relay node and the source node.

---

**Algorithm 1** Fixed Power Allocation Algorithm

---

**Initialize** Power allocation ratio $\xi^1 = 0$, step factor $\Delta\xi$, the total transmitted power of system $P_{\text{total}}$, the number of relay nodes $M$.
**Output:** The power allocation ratio$\zeta$, the sum rates $R_{\text{total}}$.
1: Initialize $s_{\max} = [1/\Delta\xi]$, $R_m^* = 0$, $R_{\text{total}} = 0$, $\zeta[]$.
2:   **for** m = $1 \rightarrow M$ **do**
3:       Determine NOMA pairing nodes $i$ and $j$ according to (11)~(13)
4:       **for** $s = 1 \rightarrow s_{max}$ **do**
5:           $\xi\# = \xi^s + \Delta\xi$
6:           Compute $a_i^m, a_j^m$ according to (14b), (14c)
7:           Compute $\Gamma_i^m, \Gamma_j^m$ according to (9) and (10)
8:           $R_{\text{sum}}^m = B/M \left( \log_2 \left( 1 + \Gamma_i^m \right) + B \log_2 \left( 1 + \Gamma_j^m \right) \right)$
9:           **if** $R_{\text{sum}}^m >= R_m^*$ **then**
10:              $R_m^* = R_{\text{sum}}^m$, $\zeta^m = \xi\#$
11:          **else**
12:              s = s + 1
13:          **end if**
14:      **end for**
15:              $R_{\text{total}} = R_{\text{total}} + R_m^*$
16:              $\zeta(m) = \xi^m$
17:  **end for**

---

### 4.2. Global Optimal Power Allocation

In this subsection, the optimization objective is set to maximize the data rate of the NOMA group while the total transmitted power is a certain value. As to the design of the sum rates-based power allocation scheme, we have referred to the optimal power control scheme proposed in Ref. [32]. It is assumed that the total transmitted power

included the transmitted powers of the transmitter and relay nodes, which also implies that the transmitted power for each NOMA group is identical. The sum rate-based power optimization problem can be formulated as:

$$\max \ R_{\text{total}} \tag{16a}$$

$$\text{subject to } 0 \ < a_i^m < a_j^m \tag{16b}$$

$$a_i^m + a_j^m \leq 1 \tag{16c}$$

$$\sum_{m=1}^{M} (P_t^m + P_m) \leq P_{total} \tag{16d}$$

$$P_t^m > 0, P_m > 0 \tag{16e}$$

It can be seen from (14d) that the power allocation optimization model of multi-user NOMA joint relay selection is a complex non-deterministic polynomial problem. In general, there is no algorithm with polynomial complexity to solve the optimal solution of this type of problem directly. Nevertheless, in the NOMA system, the pairing users share the same time–frequency resource through superposition in power domains. Additionally, since orthogonal multiple access (OMA) is applied for transmission between NOMA groups, the NOMA group consumes a certain power at each data transmission [33]. The above problem model can be equivalently simplified as follows:

$$\max \ R_m \tag{17a}$$

$$\text{subject to } 0 < a_i^m < a_j^m \tag{17b}$$

$$a_i^m + a_j^m \leq 1 \tag{17c}$$

$$P_t^m + P_m \leq \frac{P_{\text{total}}}{M} \tag{17d}$$

$$P_t^m > 0, P_m > 0 \tag{17e}$$

where $P_{\text{total}}$ takes a certain constant value. It can be easily proved that the necessary condition for the objective function to obtain the optimal solution is $a_i^m + a_j^m = 1$ and $P_t^m + P_m = P_{\text{total}}/M$. The optimization problem is demonstrated to be strictly quasi-concave with respect to $a_i^m$ and $P_t^m$. Therefore, the Lagrangian function of this optimization problem can be expressed as:

$$L_g(P_t^m, a_i^m, l_1, l_2, l_3, l_4) = -R_m + l_1(2a_i^m - 1) + l_2\left(2P_t^m - \frac{P_{\text{total}}}{M}\right) - l_3 a_i^m - l_4 P_t^m, \tag{18}$$

where $l_1$, $l_2$, $l_3$, and $l_4$ represent the Lagrange multipliers, and the Karush–Kuhn–Tucker (KKT) conditions are shown below:

$$\frac{\partial L_g}{\partial P_t^m} = -\frac{\partial R_m}{\partial P_t^m} + 2l_2 - l_4 \tag{19a}$$

$$\frac{\partial L_g}{\partial a_i^m} = -\frac{\partial R_m}{\partial P_t^m} + 2l_1 - l_3 \tag{19b}$$

$$l_1(2a_i^m - 1) = 0 \tag{19c}$$

$$l_2\left(2P_t^m - \frac{P_{\text{total}}}{M}\right) = 0 \tag{19d}$$

$$l_3 a_i^m = 0, l_4 P_t^m = 0 \tag{19e}$$

It can be calculated from (18a) and (18c) that $l_1 = 0$, and meanwhile, $l_3$ and $l_4$ are denoted by (18d), (18e) to be 0. The optimization problem is clearly shown to be a complex nonlinear optimization problem. The other parameter values are supposed to be solved by the BFGS quasi-Newton algorithm [34].

As the BFGS algorithm is used to solve the optimization problem, the iterative equation is as follows:

$$\mathbf{P}^{(s+1)} = \mathbf{P}^{(s)} + \varpi^{(s)}\mathbf{d}^{(s)}, \tag{20}$$

where $\mathbf{P} = \left(P_t^m, a_i^m, l_2\right)$, $\varpi^{(s)}$ is the step-size factor which can be defined as (20), and $\mathbf{d}^{(s)}$ is the search direction, which can be calculated by (21):

$$\varpi^{(s)} = \underset{\varpi \in \mathbb{R}}{\mathrm{argmin}} L_g(\mathbf{P}^{(s)} + \varpi\mathbf{d}^{(s)}), \tag{21}$$

$$\mathbf{d}^{(s)} = -\left(\mathbf{H}^{(s)}\right)^{-1}\left(\mathbf{g}^{(s)}\right)^{-1}. \tag{22}$$

The gradient vector of the objective function $R_m$ concerning $a_i^m$ and $P_t^m$, g, can be expressed as:

$$\mathbf{g} = \nabla F(\mathbf{P}) = \left[\frac{\partial L_g}{\partial P_t^m}, \quad \frac{\partial L_g}{\partial a_i^m}, \quad l_2\left(2P_t^m - \frac{P_{\mathrm{total}}}{M}\right)\right]^{\mathrm{T}}, \tag{23}$$

where $\mathbf{P} = \left(P_t^m, a_i^m, l_2\right)$, and Hessian matrix is as follows:

$$\mathbf{H} = \nabla F^2(\mathbf{P}) = \begin{bmatrix} \frac{\partial^2 L_g}{(\partial P_t^m)^2} & \frac{\partial^2 L_g}{\partial P_t^m \partial a_i^m} & l_2 \\ \frac{\partial^2 L_g}{\partial a_i^m \partial P_t^m} & \frac{\partial^2 L_g}{(\partial a_i^m)^2} & l_2 \\ l_2 & l_2 & 2P_t^m - \frac{P_{\mathrm{total}}}{M} \end{bmatrix}. \tag{24}$$

Thus, the iteration can be obtained by (24).

$$\nabla F(\mathbf{P}) \approx \nabla F(\mathbf{P}^{(s+1)}) + \nabla F^2(\mathbf{P}^{(s+1)})(\mathbf{P} - \mathbf{P}^{(s+1)}). \tag{25}$$

When the value of $\mathbf{P}$ is set to be $\mathbf{P}^{(s)}$, the real-valued vector calculated from the gradient vector is $\mathbf{g}_s = \nabla F(\mathbf{P}^{(s)})$, and the formulation can be made out as follows:

$$\nabla F(\mathbf{P}^{(s+1)}) - \nabla F(\mathbf{P}^{(s)}) \approx \nabla F^2(\mathbf{P}^{(s+1)})(\mathbf{P}^{(s+1)} - \mathbf{P}^{(s)}). \tag{26}$$

The BFGS algorithm takes the second-order approximation when Taylor expansion of the objective function is performed and the gradient operators on both ends of the expansion equation are applied, which simplifies the operation difficulty. For the convenience of subsequent calculations, the Hessian matrix is also approximated in the iterative process, and the correction matrix $\Delta\mathbf{B}^{(s)}$ is added, which means $\mathbf{B}^{(s)} \approx \mathbf{H}^{(s)}$, and the iterative formula can be obtained by:

$$\mathbf{B}^{(s+1)} = \mathbf{B}^{(s)} + \Delta\mathbf{B}^{(s)}, \tag{27}$$

where $\Delta\mathrm{B}^{(s)}$ is the correction matrix, which can be formulated as:

$$\Delta\mathbf{B}^{(s)} = \frac{(p^{(s)})(p^{(s)})^{\mathrm{T}}}{(p^{(s)})^{\mathrm{T}}(q^{(s)})} - \frac{\mathbf{B}^{(s)}(q^{(s)})(q^{(s)})^{\mathrm{T}}\mathbf{B}^{(s)}}{(q^{(s)})^{\mathrm{T}}\mathbf{B}^{(s)}(q^{(s)})}, \tag{28}$$

where $q^{(s)} = \mathbf{P}^{(s+1)} - \mathbf{P}^{(s)}$ and $p^{(s)} = \mathbf{g}^{(s+1)} - \mathbf{g}^{(s)}$ are the marker vector symbols.

The optimization algorithm of global optimal power allocation is designed as shown in Algorithm 2.

---

**Algorithm 2** Global Optimal Power Allocation Optimization

---

**Initialize** Target rate $R_{\text{th}}$, maximum iterations $s_{\text{max}}$, the precision threshold $\tau$, the total transmitted power of system $P_{\text{total}}$, the number of relay nodes $M$.

**Output:** The allocation of transmitted power and the Lagrange multiplier $\left(a_i^m, P_{\text{t}}^m, l_2\right)^{\text{T}}$.

1 : Set the initial power allocation coefficient and transmitted power $\left(a_i^{m^*}, P_{\text{t}}^{m^*}, l_2\right)^{\text{T}} = \left(1/2, P_{\text{total}}/2M, 0\right)^{\text{T}}$.

2: **for** $s = 1 \rightarrow s_{max}$ **do**

3: 　　Determine the search direction

4: 　　　　$\mathbf{d}^{(s)} = -(\mathbf{B}^{(s)})^{-1} \times \mathbf{g}^{(s)}$

5: 　　　Compute the step factor $\varpi^{(s)}$ according to (20)

6: 　　　Compute the mark vector

7: 　　　　$q^{(s)} = l^{(s)}\mathbf{d}^{(s)}, \mathbf{P}^{(s+1)} = \mathbf{P}^{(s)} + q^{(s)}$

6: 　　　Compute $||\mathbf{g}^{(s+1)}||_2$

7: 　　　　**if** $||\mathbf{g}^{(s+1)}||_2 < \tau$ **then**

8: 　　　　　　**break**

9 : 　**else** compute $\Gamma_i^m, \Gamma_j^m$ according to (9) and (10)

10 : 　　$R_i^m = B/M\left(\log_2\left(1 + \Gamma_i^m\right)\right), R_j^m = B/M\left(\log_2\left(1 + \Gamma_j^m\right)\right)$

11 : 　　**if** $R_i^m > R_j^m \& R_j^m >= R_{\text{th}}$ **then**

12: 　　　　　　Compute $p^{(s)} = \mathbf{g}^{(s+1)} - \mathbf{g}^{(s)}$ according to (25)

13: 　　　　　　Compute $\mathbf{B}^{(s+1)} = \mathbf{B}^{(s)} + \Delta\mathbf{B}^{(s)}$ according to (27)

14: 　　　　　　Let $s = s + 1$ and return to step 3

15: 　　　　　**else break**

16: 　　　　**end if**

17: 　　　**end if**

18: **end for**

---

### 4.3. Stepwise Sub-Optimization Power Allocation

The optimal solution for the system can be obtained by our proposed GOPA algorithm, proposed in Section 4.1, through traversing the overall system. Nonetheless, the complexity of the system algorithm will be raised to a great degree as the number of nodes accessing the network increases and hence, we further propose a step-by-step sub-optimization algorithm (SSOPA) with a lower computing complexity.

In this problem formulation, the power allocation coefficients in the NOMA group are preferentially determined, with the relay transmitted power assumed to fulfil the QoS (quality of service) of the edge user in the NOMA group. Under this condition, the power allocation issues can be optimized between the transmitter and relay nodes. In particular, the CSIs between the transmitter and the relays, and between the relays and receiving nodes, are comprehensively considered. According to (1), the power allocation coefficients can be conducted as follows:

$$\begin{cases} a_i^m = \dfrac{1}{1 + \left(\left(h_i h_{m,i}\right)/\left(h_j h_{m,j}\right)\right)} \\ a_j^m = \dfrac{\left(h_i h_{m,i}\right)/\left(h_j h_{m,j}\right)}{1 + \left(\left(h_i h_{m,i}\right)/\left(h_j h_{m,j}\right)\right)} \end{cases}. \tag{29}$$

The problem model can be simplified as follows:

$$\max \ R_m \tag{30a}$$

$$\text{subject to } P_{\text{t}}^m + P_m \leq P_{\text{total}}/M \tag{30b}$$

$$P_{\text{t}}^m > 0, P_m > 0 \tag{30c}$$

This optimization model is still an optimizing problem of nonlinear functions. To reduce the complexity of the algorithm, the particle swarm algorithm (PSA) is employed to solve this problem, which is easier to be implemented. As a bionic algorithm, the PSA has natural advantages in solving global optimization problems. Compared with the genetic

algorithm, the PSA can memorize all solutions, only needs to update according to the internal speed, and its convergence speed is even faster [35].

The fitness function in this problem is defined as follows:

$$\text{fit}(P_t^m, P_m) = R_m. \tag{31}$$

The PSA's solution procedure is as follows:

(1) The particle swarm is initialized randomly. Each particle carries two types of information: position information $c$ and velocity information $V$, with the position information corresponding to the value taken in the fit function, which is the allocated transmitted power $[P_t^m, P_m]$. We define the maximum number of iterations $s_{\max}$, the particle swarm size $K$, and the particle's maximum speed $V_{\max}$.

(2) The initial adaptation value of the particles in each population is calculated according to (30) to obtain the optimal solution *pBest* in that population.

(3) Searching the initial optimal solution *gBest* in all swarms.

(4) Update particle velocity and position. During the $t$th iteration of the $k$th particle, the value of $c_k$ and the speed value of $v_k$, are updated by:

$$v_k^{(t)} = \chi v_k^{(t-1)} + \vartheta_1 r_1 \left( \boldsymbol{pbest}_k - \boldsymbol{c}_k^{(t-1)} \right) + \vartheta_2 r_2 (\boldsymbol{gbest}^{(t-1)} - \boldsymbol{c}_k^{(t-1)}), \tag{32}$$

$$\boldsymbol{c}_k^{(t)} = \boldsymbol{c}_k^{(t-1)} + \boldsymbol{c}_k^{(t-1)}, \tag{33}$$

where $\chi$ represents the inertia weight. When $\chi$ becomes larger, the global optimizing ability of the model is enhanced, and the local optimizing ability is weakened. $c_1$ and $c_2$ denote acceleration coefficients that reflect the intensities of particle self-learning and group learning.

(5) Return to step 2.

The optimizing algorithm for the global optimal power allocation is designed as shown in Algorithm 3.

---

**Algorithm 3** Stepwise Sub-Optimization Power Allocation

---

**Step 1** Compute the power allocation coefficient of pairing nodes according to (26)
**Step 2** Optimize the power allocation between transmitter and relay node
**Initialize** The number of particle swarm $K$, the maximum iteration $s_{\max}$, the initial adaptive functionvalue $R_m^* \leftarrow 0$.

**Output:** The allocation of transmitted power$(P_t^m, P_m)^{\mathrm{T}}$.
1:     **for** $k \leftarrow 1$ **to** $K$
2:         Initialize the initial source node transmission power factor $\boldsymbol{c}_k$ and the velocity factor $\boldsymbol{v}_k$.
3 :   compute $\Gamma_i^{m,k}, \Gamma_j^{m,k}$ according to (9) and (10)
4 :     $R_m^{(k)} = B/M \left( \log_2 \left( 1 + \Gamma_i^{m,k} \right) + \log_2 \left( 1 + \Gamma_j^{m,k} \right) \right)$
5 :     **if** $R_m^{(k)} > R_m^*$ **then**
6:             $\boldsymbol{pbest}_{(k)} \leftarrow \boldsymbol{c}_{(k)}, \boldsymbol{gbest} \leftarrow \boldsymbol{c}_{(k)}, R_m^* \leftarrow R_m^{(k)}$
7:         **else break**
8:     **end if**
9:   **end for**
10:       **while** not **stop**
11:           **for**$s \leftarrow 1$ to $s_{\max}$
12:           Update the source node transmission power factor $\boldsymbol{c}_{(k)}$ and the velocity factor $\boldsymbol{v}_{(k)}$ according to (31) and (32)
13:               **if** fit($\boldsymbol{c}_{(k)}$) > fit($\boldsymbol{pbest}_{(k)}$) **then**
14:                   $\boldsymbol{pbest}_{(k)} \leftarrow \boldsymbol{c}_{(k)}$
15:               **if** fit($\boldsymbol{pbest}_{(k)}$) > fit($\boldsymbol{gbest}$) **then**
16:                   $\boldsymbol{gbest} \leftarrow \boldsymbol{pbest}_{(k)}$
17:               **end for**
18 :     $[P_t^m, P_m] \leftarrow \boldsymbol{gbest}$
19:       **end while**

---

### 4.4. Analysis of Algorithm Complexity

In this paper, three algorithms are proposed to solve the problem model in different ways. This section analyzes the three algorithms in terms of time complexity and problem size, in order to measure the time and resources consumed by the three algorithms during the calculation process.

The FPA method can complete the algorithm's solution with a simple nested loop, and the sum rate can be solved in the nested problem using simple multiplication and addition operations. The matrix multiplication operation is required in GOPA. Although we optimize the search direction and step factor solution process by constructing an approximate Hesse matrix rather than solving the Hesse process, the problem scale remains relatively large at this time. The algorithm complexity of SSOPA is proportional to the number of iterations and the initialized population size, and the problem size in a single iteration is small. The problem size of a single mathematical calculation is set to 1 here. To summarize, the complexity statistics for the three algorithms are shown in Table 2. Some of the variables employed in this paper are summarized in Table 3.

**Table 2.** Comparison of Algorithm Complexity.

| Algorithm | Time Complexity | Problem Size |
|:---:|:---:|:---:|
| FPA | $O(s_{\max})$ | $107M$ |
| GOPA | $O(s_{\max}^2)$ | $1570M/\tau$ |
| SSOPA | $O(s_{\max})$ | $278MK$ |

**Table 3.** Variables commonly used in this paper.

| | | | |
|:---|:---|:---|:---|
| x | Transmitted signal | y | Received signal |
| P | Power | h | Channel gain |
| n | Number of the receiving nodes, $n \in [1, N]$ | m | Number of the relay nodes, $m \in [1, M]$ |
| a | NOMA power allocation coefficient | Z | Power allocation ratio |
| $\bar{n}$ | Noise power | $s_{\max}$ | Iterations of algorithm |
| $\tau$ | Step-size factor | k | Number of the particle swarm, $k \in [1, K]$ |
| $\delta$ | Amplification coefficient | d | Transmission distance |
| c | Attenuation coefficient | $\theta$ | Beam divergence angle |
| $\theta_0$ | Inclination angle | $\alpha$ | Optical fading amplitude |

## 5. Simulation and Analysis

In this section, we will present comprehensive simulation studies to verify the effectiveness of the proposed schemes. In this paper, we mainly consider the scenarios for the line-of-sight communication links. The configuration scheme of the transmitter in this paper refers to the quasi-omnidirectional prismatic LED array module that is proposed in [36]. The parameter settings are summarized in Table 4.

**Table 4.** Simulation parameters.

| Parameters | Values |
|:---:|:---:|
| System bandwidth $B$ | 32 MHz |
| Total transmitted power $P_{\text{total}}$ | 2000 mW |
| Divergence angle of the transmitter $\theta_0$ | 30° |
| Aperture area of the optical receiver $A_{\text{de}}$ | 0.01 m$^2$ |
| P-I modulation conversion coefficient $\eta_{\text{tr}}$ | 0.9 A/W |
| Responsibility of the photodetector $\eta_{\text{de}}$ | 0.9 W/A |

### 5.1. Simulation Scenarios for NOMA–UOWC Network

To study the performance of the relay-aided NOMA–UOWC system, three scenarios are designed as shown in Figure 2. For the convenience of simulation calculation, a rectangular coordinate is established with the source node as the origin. In scenarios 1 and 2, the specific positions of receiving nodes are given as (8.69, 2.33), (11.59, −3.11), (14.94, −1.31), and (17.93,1.57), severally, and the Euclidean distances between the source node and receiving nodes are 9 m, 12 m, 15 m, and 18 m, separately. The tilt angles are set up as 15°, −15°, −5°, and 5°, respectively. In scenario 3, the tilt angle of the receiving nodes remains unchanged, but the Euclidean distances between the source node and receiving nodes increase to 18 m, 24 m, 30 m, and 36 m, separately. During the deployment phase of the relay nodes, we refer to the relay deployment method for single-hop networks of Ref. [17], which has been mentioned in the introduction, and temporarily disregard the geometric loss of the optical communication link. The specific positions of relay nodes are given as (3.94, 0.69) and (6.89, −1.22) in scenarios 1 and 2, while the tilt angles are set up as 10° and −10°. As with the deployment of receiving nodes, in scenario 3 the tilt angle of relay nodes remains unchanged, and the Euclidean distances between the source node and relay nodes increase to 8 m and 14 m, respectively. The specific position of each node is shown in Figure 2.

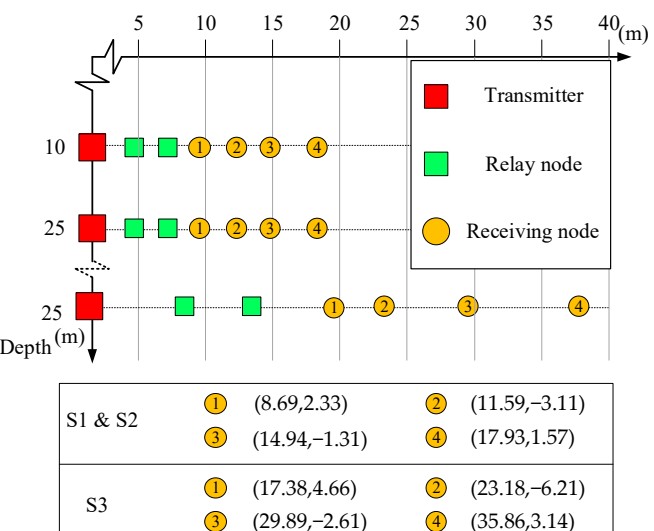

**Figure 2.** Diagram of simulation scenarios.

In this subsection, our main work is to verify the rationality of the proposed system model, and the simulations are performed temporarily, considering only a single NOMA group transmission. The transmission power only occupied half of the total system power. Prior to further system performance analysis, the effectiveness of NOMA for underwater wireless optical communication systems is to be validated. In this case, a comparative simulation is performed to investigate the performances of NOMA-based UOWC and OMA-based UOWC with the same available bandwidth resources, and the receiving nodes in the transmission model are opted for node 1 and node 3 of scenario 1 in Figure 2. The most current $2 \times 2$ NOMA network is applied in this simulation. The detailed results can be seen in Figure 3. We can easily conclude from Figure 3 that the sum rate of the system increases when the power transmitted increases. It is notable that a NOMA-based system can support almost 1.4 times more effective communication sum rates than an OMA-based systems for the UOWC network when the transmitted power is 1000 mW, due to sharing the same time–frequency resource.

One more important issue that we consider for the research of NOMA–UOWC network is the user pairing. To demonstrate the system performance variation of a three-user paired NOMA relative to a traditional two-user paired NOMA with the same bandwidth and

power resources, a simple simulation is conducted with the results provided in Figure 4. Referring to the theoretical analysis in Section 3, the total transmitted power in the two simulations are assumed to be the same, 1000 mW. The transmission distances of user 1 and user 2 are 4 m and 9 m, respectively. The blue line is a control group presented in the simulation, showing the system sum rates of the traditional two-user paired NOMA. In other words, the resources allocated to the third user are zero, so the system sum rates are constant. The orange line shows the system sum rates for three-user paired NOMA. Compared to the control group, it includes the original two-user and the newly increased user 3. As the Euclidean distance between the newly increased user 3 and the transmitter increases, the system sum rates decrease. Furthermore, we can obtain the following conclusions. As the number of paired users increases in a single NOMA group, the system sum rates do not definitely increase for a constant power resource, and the sum rates of three-user paired NOMA are better than that of two-user paired NOMA only when user 3 is the nearest user to the transmitter. Hence, we mainly consider the $2 \times 2$ NOMA networks for the UOWC network based on cooperative transmission in this paper.

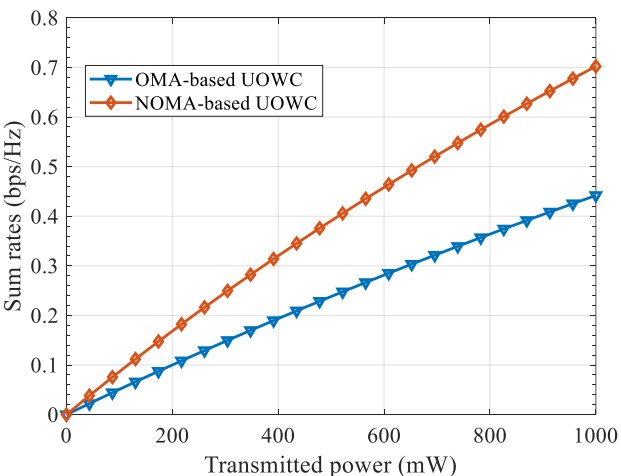

**Figure 3.** Sum rates versus transmitted power for OMA-based UOWC and NOMA-based UOWC.

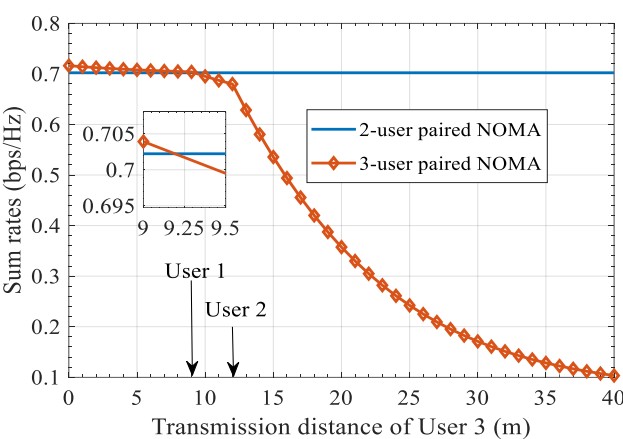

**Figure 4.** Sum rates of relay-aided UOWC system with two-user paired and three-user paired NOMA.

In addition, to highlight the performance of the relay-aided NOMA scheme (denoted as ICNOMA), ICNOMA is compared with two different schemes, which are summarized as follows. (a) ONOMA: a NOMA system with no relay or cooperative communication adopted, seen in Figure 5a. (b) CNOMA: it is assumed that the near-end node has both signal reception and signal forwarding functions in the NOMA system shown in Figure 5b.

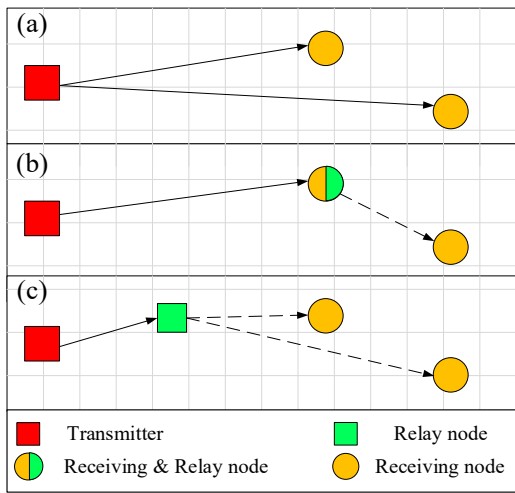

**Figure 5.** Diagram of three kinds of NOMA schemes. (**a**) Ordinary NOMA, (**b**) cooperative NOMA without independent relay nodes, (**c**) independent-relay-aided NOMA.

### 5.2. Performance Evaluations

To verify the effectiveness of the proposed relay-aided NOMA under different user pairing conditions, relative to the NOMA scheme proposed in Ref. [37] and the conventional NOMA scheme, a series of simulations were performed, and the detailed results are presented in Figure 6. In Figure 6, 1&2, 1&3, and 1&4 represent user 1 and user 2, 3, and 4 pairings, respectively. The system model of each NOMA scheme can be referred to in Figure 5. To minimize the interference of other components, the simulation environment is set as scenario 1, and the relay node 1 is set to be the relay node. Since the relay nodes have been identified, the FPA algorithm is performed in both NOMA schemes. The results for ONOMA, CNOMA, and ICNOMA are shown in Figure 6. It is evident that the introduction of cooperative communication can significantly improve the performance of a NOMA system. Therein, an independent relay can further enhance the system sum rate; for instance, the sum rates for ICNOMA with different users paired are enhanced by 0.6 bps/Hz to 0.9 bps/Hz compared to the CNOMA when the transmitted power is the same, 1000 mW.

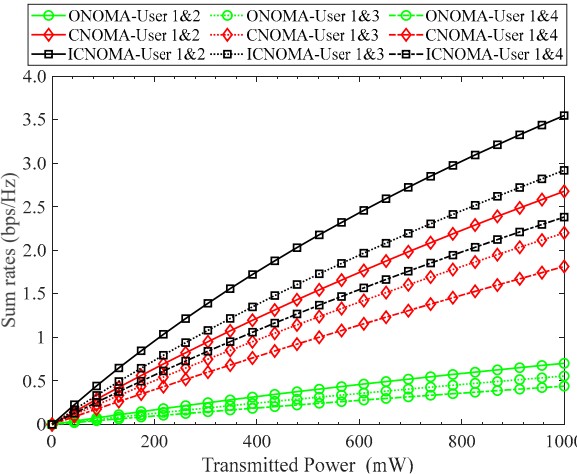

**Figure 6.** Sum rates versus transmitted power of NOMA-based UOWC system.

For the purpose of assessing the proposed GOPA and SSOPA algorithms, the sum rates of the NOMA–UOWC network for three scenarios are illustrated in Table 5, where the FPA represents a fixed power allocation scheme, in which the transmitted power for the transmitter and relay nodes are evenly distributed. The attenuation coefficient

of seawater is selected as 0.056 m$^{-1}$ to analyze the impact of solar noise on the UOWC network performance. Therein, the GOPA and SSOPA algorithms perform the prominent ability in improving the system communication capability in scenario 1. Significantly, these two proposed schemes have the absolute advantage over the traditional FPA method, due to the fact that the proposed user pairing scheme jointed with relay selection not only optimizes the cooperative transmission, but also equilibrates the channel fading caused by the path loss and the ocean turbulence. It is noted that in scenario 3, with the increases in depth of nodes and the transmission distance, the GOPA algorithm has a better capability of balancing the path loss and the solar noise. It can be concluded that the GOPA algorithm presents the ability to extend the communication range of the UOWC network.

**Table 5.** Comparisons of three scenarios for ICNOMA–UOWC.

| | | Relay Selection | Relay Selection | $R_{total}$ (bps/Hz) |
|---|---|---|---|---|
| | GOPA | $C_1$ ($i = 1, j = 4$) | $C_2$($i = 2, j = 3$) | 7.92 |
| Scenario 1 | SSOPA | $C_1$ ($i = 1, j = 4$) | $C_2$($i = 2, j = 3$) | 7.62 |
| | FPA | $C_2$ ($i = 1, j = 3$) | $C_2$($i = 2, j = 4$) | 6.10 |
| | GOPA | $C_1$ ($i = 1, j = 4$) | $C_2$($i = 2, j = 3$) | 11.05 |
| Scenario 2 | SSOPA | $C_1$ ($i = 1, j = 4$) | $C_2$($i = 2, j = 3$) | 10.87 |
| | FPA | $C_1$ ($i = 1, j = 3$) | $C_2$($i = 2, j = 4$) | 8.65 |
| | GOPA | $C_1$ ($i = 1, j = 4$) | $C_1$ ($i = 2, j = 3$) | 2.15 |
| Scenario 3 | SSOPA | $C_1$ ($i = 1, j = 4$) | $C_1$ ($i = 2, j = 3$) | 1.50 |
| | FPA | $C_1$ ($i = 1, j = 3$) | $C_1$ ($i = 2, j = 4$) | 0.37 |

The results of the relay selection and power allocation for the proposed network in the clear seawater of scenario 1 are summarized in Table 6. To analyze the impact of cooperative transmission on the UOWC network performance, a control group without a cooperative relay was introduced in the performance evaluation. It was noted that the optimal results can be obtained by the GOPA algorithm; meanwhile, the SSOPA algorithm achieves a sub-optimal solution. It can be observed that the GOPA algorithm increases the sum rate by 0.034 bps/Hz in comparison with the SSOPA algorithm. To further evaluate the impact of cooperative transmission on UOWC network performance, the None-relay (a control group without a cooperative relay) scheme was performed [18]. The results of the four schemes show that the relay-assisted approaches prove to be distinct superior compared to the None-relay mode, which verifies the effectiveness of the proposed relay-aided NOMA network. In addition, the GOPA algorithm and SSOPA algorithm proposed in this paper retain the effectiveness in improving the system sum-rates in a conventional NOMA system without cooperative communication.

For the purpose of assessing the proposed scheme in different ocean environments, the comparisons of GOPA, SSOPA, and FPA algorithms are demonstrated in Figure 7, where the scintillation indexes reflect the turbulence intensity. We adopt the FPA algorithm as a contrast to assess the proposed GOPA and SSOPA algorithms. Among these three methods, the GOPA algorithm has the most obvious sum-rate increase compared to the others. Therein, the increment shows obvious changes with different $\sigma_{sci}^2$ varying from 0.1 to 1.0. This is due to the fact that the power allocation efficiency has disparate computation complexity in different turbulent environments. It can be summarized from Figure 7a that the sum rate of the GOPA algorithm is nearly the same as that of the SSOPA algorithm, with $\sigma_{sci}^2 = 0.1$. Meanwhile, the sum rate of the GOPA algorithm exceeds those of SSOPA and FPA algorithms by far, with $\sigma_{sci}^2 = 1.0$. We can conclude that the sum rate of GOPA is clearly improved in the three scenarios, and the SSOPA algorithm shows relatively better performance in the weaker turbulence environment.

**Table 6.** Comparisons of three schemes for ICNOMA–UOWC.

| | Relay Selection | $P_{t_m}$ (mW) | $P_m$ (mW) | $a_{i_m}$ | $a_{j_m}$ | $R_{\text{total}}$ (bps/Hz) |
|---|---|---|---|---|---|---|
| GOPA | $C_1$ ($i = 1, j = 4$) | 421.52 | 578.48 | 0.206 | 0.796 | 7.92 |
| | $C_2$ ($i = 2, j = 3$) | 361.20 | 638.80 | 0.377 | 0.623 | |
| SSOPA | $C_1$ ($i = 1, j = 4$) | 445.73 | 554.27 | 0.211 | 0.789 | 7.62 |
| | $C_2$ ($i = 2, j = 3$) | 356.85 | 643.15 | 0.384 | 0.616 | |
| FPA | $C_1$ ($i = 1, j = 4$) | 500.00 | 500.00 | 0.191 | 0.809 | 6.10 |
| | $C_2$ ($i = 2, j = 3$) | 500.00 | 500.00 | 0.401 | 0.599 | |
| N-relay with GOPA | ($i = 1, j = 4$) | 1000.00 | 0.00 | 0.051 | 0.949 | 1.08 |
| | ($i = 2, j = 3$) | 1000.00 | 0.00 | 0.113 | 0.887 | |
| N-relay with SSOPA | ($i = 1, j = 4$) | 1000.00 | 0.00 | 0.045 | 0.955 | 0.97 |
| | ($i = 2, j = 3$) | 1000.00 | 0.00 | 0.101 | 0.899 | |
| N-relay with FPA | ($i = 1, j = 4$) | 1000.00 | 0.00 | 0.034 | 0.966 | 0.74 |
| | ($i = 2, j = 3$) | 1000.00 | 0.00 | 0.087 | 0.913 | |

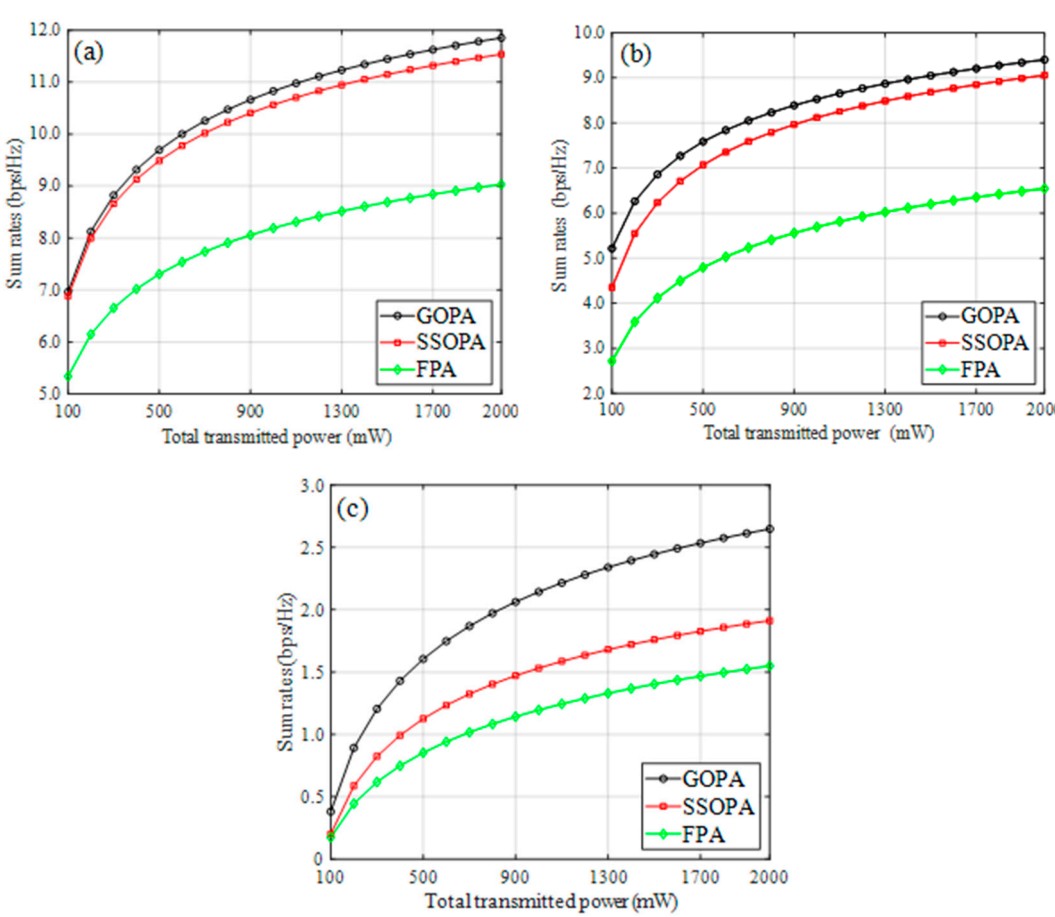

**Figure 7.** Sum rates versus transmitted power of ICNOMA–UOWC system for (**a**) $\sigma^2_{\text{sci}} = 0.1$, (**b**) $\sigma^2_{\text{sci}} = 0.4$, (**c**) $\sigma^2_{\text{sci}} = 1.0$.

To further study the impact of solar noise on the results of a relay-aided NOMA–UOWC network, three typical ocean seawaters, which can be referred to in Table 1, were employed to create the depth simulation. It can be observed that the receiving sum rates of NOMA nodes change greatly with the node depth varying from 0 m to 50 m. Significantly, the receiving data rates in the harbor water are far too low compared to the other three conditions. Therefore, the comparisons of these three scenarios are shown in Figure 8. Similar to the results detailed above, the GOPA algorithm obtains the optimized

performance in the sum rate of the NOMA–UOWC network under all conditions. However, the gap between GOPA and SSOPA algorithms has narrowed almost to vanishing point in the clear ocean water. In contrast, the GOPA algorithm shows a considerable advantage to the SSOPA and FPA algorithms in the coastal water, with the sum rate of the proposed network showing a maximum at the depth of around 20 m. In the meantime, the sum rates of the proposed network in clear ocean water and pure ocean water increase approximately in line with the depth within 50 m. According to the formulas derived above, the solar noise is positively associated with the depth of node, from which we can deduce that the depth of node has a crucial impact on the performance of the UOWC network. However, the impact of solar noise in the coastal water is relatively weak, which is caused by the fact that the path loss and the noise are balanced at a depth of around 20 m. From a comprehensive perspective, the proposed algorithms of relay selection and power allocation optimization have outstanding capabilities of not only balancing the path loss and solar noise, but also equilibrating the channel fading between the transmitter and the relays, and between the relays and the receiving nodes.

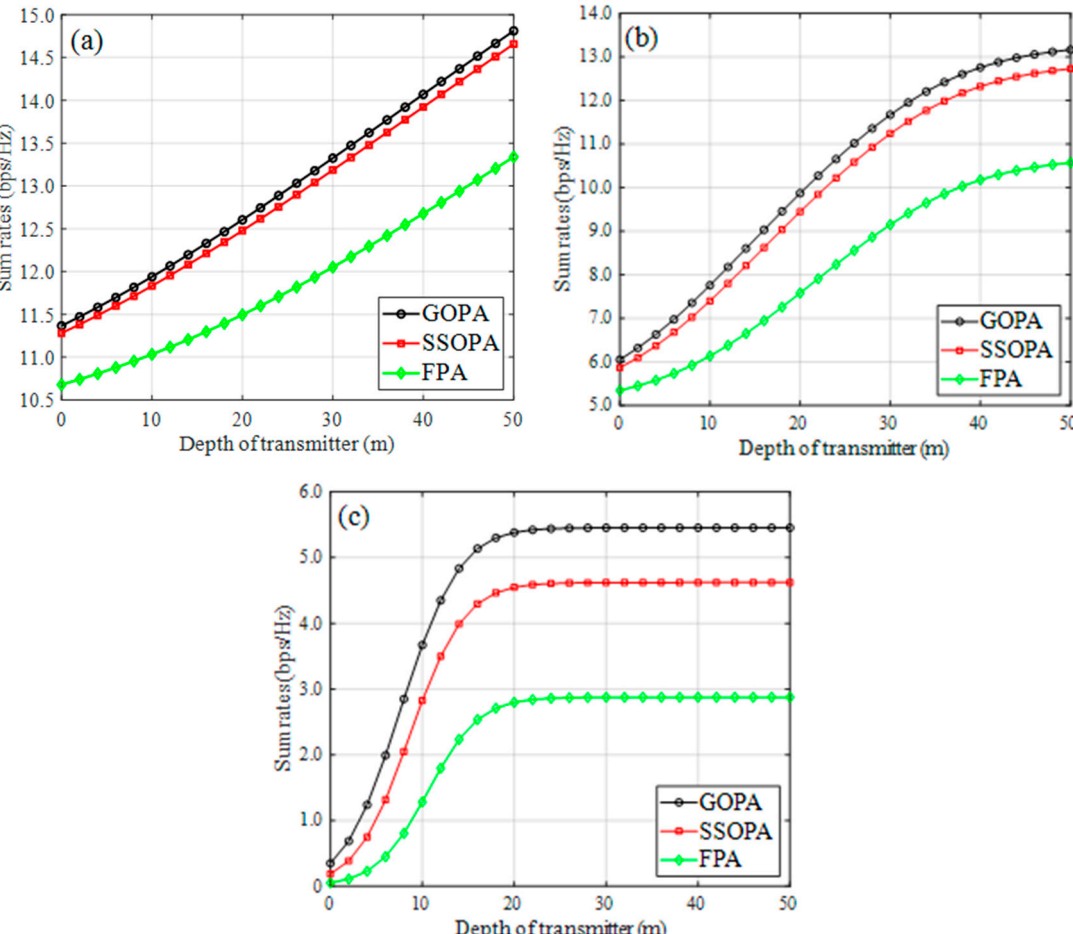

**Figure 8.** Sum rates versus depth of transmitter for ICNOMA system for: (**a**) clear ocean water, (**b**) pure ocean water, and (**c**) coastal water.

## 6. Conclusions

A relay-aided NOMA scheme for UOWC networks is proposed in this paper, in which a user pairing method of joint relay selection is suggested. Due to solar noise, two power allocation-optimizing algorithms based on the maximization of the sum rates, i.e., the GOPA and SSOPA algorithms, are designed. Both schemes are proven to be quasi-convex optimizing problems, whose solutions are obtained by the BFGS Newton algorithm and the particle swarm algorithm, respectively. The proposed algorithms are evaluated in

AF-based cooperative transmission UOWC networks, and the simulation results verify that the proposed optimization scheme performs more effectively in improving the system communication capability. Furthermore, solar radiation noise has been shown to be one of the main factors affecting NOMA–UOWC network performance.

**Author Contributions:** Conceptualization, Y.L., H.Y.; formal analysis, Y.L.; methodology, Y.L. and H.Y.; software, Y.L., L.J., X.J. and J.W.; validation, Y.L., X.J. and J.W.; writing—original draft, Y.L.; writing— review and editing, H.Y. and Y.L.; visualization, Y.L.; supervision, H.Y.; project administration, H.Y.; funding acquisition, H.Y. All authors have read and agreed to the published version of the manuscript.

**Funding:** This work is supported in part by the National Natural Science Foundation of China (NSFC) under Grant 61871418 and 61801079, China.

**Data Availability Statement:** The data presented in this paper are available after contacting the corresponding author.

**Acknowledgments:** The authors would like to thank the anonymous reviewers for their careful reading and valuable comments.

**Conflicts of Interest:** The authors declare no conflict of interest.

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
