# Peer review of "Performance Analysis of Relay-Aided NOMA Optical Wireless Communication System in Underwater Turbulence Environment"

_remotesensing, doi:10.3390/rs14163894_

Round 1

Reviewer 1 Report

I am the second reviewer of the previous submission. I would like to thank the authors for the detailed answer to my comments. I have a quick scan, and feel the paper indeed offers some good information, but the authors do not clearly illustrate the analysis conditions in an organized manner. Below are some concerns.

1.     For Fig. 6., is the variable of the x-axis “SNR” or “Power”? I assume it is a typo.

2.     For Fig. 3 and Fig. 4, the spectrum efficiency is both less than 1 bps/Hz, but after Fig. 6, the spectrum efficiency is mostly much higher than that values. It would be helpful if the authors could explain more.

3.     If I understand correctly, Fig. 4 indicates the system has identical spectrum efficiency for all transmission distance? For the transmitter divergence angle as large as 30 deg. indicated in Table 4, shouldn’t the received power drops quickly with distance, and results in varied SNR? Could the authors explain why the spectrum efficiency is not influenced by the distance?

4.     I am not sure about the significance of the comparison for Fig. 6. It is understood that the authors try to emphasize the performance benefit of the relay aided scheme. However, as more as I understood, the system performance is greatly determined by factors like the optic sub-system and the user spatial distribution. Unless the authors identify the operation conditions valid for Fig. 6, I feel Fig. 6 only stands for a special case. It would be helpful if the authors can illustrate how it is a representative case for the system performance. For my previous question about why the authors get much better spectrum efficiency compared to Ref. [B], I think the authors just need to clarify for the value as high as 11 bps/Hz is necessary. This is the same as my comment 2 given above.

5.     About the authors’ answer for the unnecessity of guard mechanisms to perform OMA, I guess the authors mean that each NOMA group uses its dedicated frequency channel or time slot? Then unless an infinite total system bandwidth is available, should the channel bandwidth available for each NOMA group decrease with the total number of groups? Should this (besides the power) also be added into the optimization criteria? It would also be helpful if the optimization criteria for the calculation of the OMA case is explicitly stated.

6.     As more as I understand, this is purely a simulation work. I suggest the authors avoid using the wording of “experiment.”

Author Response

Dear reviewer:

     Thank you for your decision and constructive comments on my manuscript. We have carefully considered the suggestion of Reviewer and make some changes. We have tried our best to improve and made some changes in the manuscript.

     The blue part that has been revised according to your comments. Revision notes, point-to-point, are uploaded as a Word file.

Reviewer 2 Report

Fusion of NOMA and UOWC is an interesting part of this paper. However, I have many major concerns as given below,

-If any term used many times or more than once throughout the paper, then we generally defined acronym for any term in 1st appearance

and it is must to use that acronmy in rest of the paper.

-In conclusion: underwater optical wireless networks--> UOWC.

-Many Undefined acronyms in 1st appearance,

e.g., BFGS, LED etc.

-FPA (fixed power allocation)-->fixed power allocation (FPA)

-Punctuation errors to be taken care

-Remove dots from all the equations, if they do not indicate dot products.

-All equations much be properly ended up with `.' or `,'.

-Provide a table for the notations.

-Symbol used to represent same parameter in fig 1 and fig 2 are different!!!! do the needful! Besides, Why is receiving nodes colour different in fig2(a) and fig2(b)?

-Equations 1 and 5 are important part of your analysis. But I think eqs.(3) and (4) are erroneous as far as NOMA is concerned. Recheck and update as deemed fit.

Below reference is useful to fix the concern,

Towards a 20 Gbps multi-user bubble turbulent NOMA UOWC system with green and blue polarization multiplexing, Optics Express Vol. 28, Issue 21, pp. 31796-31807 (2020)

-Cite the following papers in order to improve the introduction and the mathematical part,

Underwater Optical Wireless Communication, IEEE access, vol.4, pp.1518-1547.

Underwater Optical Wireless Communications: Overview, Sensors 2020, 20(8), 2261; https://doi.org/10.3390/s20082261

A Novel Channel Model and Optimal Power Control Schemes for Mobile mmWave Two-Tier Networks, IEEE access, vol.10, pp.54445 - 54458.

Underwater optical wireless communications, networking, and localization: A survey, Ad Hoc Networks, vol 94, November 2019, 101935.

-The results to be clearly presented.

-Thorough proofread requires.

Author Response

(The authors gave the same response as above.)

Reviewer 3 Report

Using general non-orthogonal multiple access (NOMA) pairing node methods, the authors suggested two novel power optimization schemes for NOMA underwater optical wireless communications (UOWC). A global optimal power allocation algorithm and a stepwise sub-optimization power allocation algorithm are designed for the relay-assisted NOMA communication network based on amplifying-and-forwarding. The simulation results proved the effectiveness of the proposed schemes. Based on my knowledge the results are novel and deserve to be published.

Round 2

Reviewer 1 Report

I feel the authors try to divide their paper into many different focusing points, including verification of the model, showing their performance improvement, the benefits from the other node connection schemes, etc. However, different reference criteria/presumptions are applied for different discussion points of the analyses. Hence, it is not easy (at least for me) to see the main benefits this paper may offer. (It may be reasonable to guess that the system performance is not as good as claimed while different criteria/presumptions are applied, so the authors try to use different criteria/presumptions at different points). I suggest the authors unify their reference criteria/presumptions for all points of the whole paper, and discuss accordingly. I do not plan to comment in further details for an extra run. Considering the fact that this paper indeed may offer some good ideas to the readers and the positive comments given by the other reviewers, I suggest the paper be accepted.

Reviewer 2 Report

No more comments, the present form of the paper is now ready to accept.

This manuscript is a resubmission of an earlier submission. The following is a list of the peer review reports and author responses from that submission.

Round 1

Reviewer 1 Report

The authors study relay selection and power allocation of the relay assisted NOMA underwater optical wireless communication network. They combine NOMA and cooperative communications technology. They suggest user pairing strategy for joint relay selection. Two algorithms are presented by authors for optimal power allocation. Sum rates are presented versus transmitted power and versus depth of transmitter. Some interesting conclusions are emphasized.

To the best of the Reviewer’s knowledge the results are novel. The paper is written well and can be followed. The paper is based on optimization procedures, and general problem of these type of papers is that it is not easy to check the results. However, it seems that everything is correct. Except one typo on page 4, line 129, “Pm” should be replaced by “P_m”, this Reviewer has not any other important suggestions and think that the paper could be accepted for publication in the present form.

Author Response

Response 1: Thank you very much for your meticulous pointing out the problem for us and we have modified it.

Reviewer 2 Report

This paper studies a relay-aided UOWC system. I think the idea is interesting. However, though I have merely a quick scan, I have some concerns as below:

  1. I may miss some details for Fig. 2. However, as more as I see, I think the scenarios listed in Fig. 2 is not practical. As has already been included in Table 2 of the manuscript, the beam angle is a significant factor for the communication system. I believe the spatial distribution (at least 2D over the plane) of different nodes should be considered. Some receiving nodes outside the beam footprint cannot receive the beam from the relay even if they are within the distance considered by the authors.
  2. Bandwidth as low as 20 MHz is considered for the simulation. Considering lots of high speed experiment results have been demonstrated, I wonder why the authors choose this value for simulation. Please explain. By the way, the spectrum efficiency as high as 11 b/s/Hz is calculated. Could the authors comment this value according to other published experimental results?
  3. Only two nodes are sharing each relay? Why not more? Could the authors comment under what conditions, the NOMA scheme over two nodes may be advantageous?
  4. I suggest the authors can discuss their contribution in regard of practical system performance (may be compared with some published works). For the current content, I cannot easily catch the key contributions of the paper. The studies for relay-aided scheme are interesting, but not as novel as could be the key factor for this paper to be accepted. The simulation results also seem not appropriately explain for practical scenarios.

Author Response

Thank you so much for your constructive opinion. In the attachment, we provide a point-by-point response to your comments.

Reviewer 3 Report

Pl find the following comments in order to improve the quality of the paper,

-many undefined acronyms in its 1st appearance, e.g., 
BFGS, OFDMA, FPA etc.

-remove dots from (1), (24) etc.

-provide complexity analysis for both algorithm 1 and algorithm 2. 

-n is used to denote both `n^th receiving node' and `noise'. To avoid confusion use different notations.
Hence, provide a table for all notations.

-how are two optimization schemes being solved by the BFGS algorithm and the particle swarm algorithm, respectively? needs more clarity!!!

-FPA algorithm has not been discussed anywhere in the paper, but all of a sudden it appears in all the simulation results? surprising for me!!!!

-introduction could have been better?

-cite few recent references (e.g., 2022, 2021, 2020) from MDPI venues.

Author Response

(The authors gave the same response as above.)

Round 2

Reviewer 2 Report

I am the second reviewer of the previous run. I would like to thank the authors for the detailed report. I quickly overview the response report, and my thoughts are as below:

1.      The horizontally separated scenarios are sufficient for me. However, I suggest the authors clearly labeled the spatial information of the horizontal planes. This will more clearly reveal under what conditions the results are calculated for.

2.      I understand the authors try to emphasize that they get much higher spectral efficiency than Ref. [B] in the response report. However, I feel confused that what may be the reasons for the authors to get almost double spectrum efficiency. The “theoretical limit” is irrelevant to the algorithms applied. As mentioned by the authors, this manuscript follows the basic analysis scheme of Ref. [B], and further considers the solar interference. Then, shouldn’t the “optimal” condition of Ref. [B] be even better than the “GOPA” condition of this manuscript?

3.      It is interesting that the authors conclude that 2-user NOMA is sufficient. I believe this could be true but controversial. I understand it is difficult to consider all possible conditions. However, I suggest the authors may at least comment under what conditions (users’ spatial distribution, number of users, total power, etc.) their conclusions may apply. I also suggest the authors may add one picture showing the system performance while no NOMA is applied. By the way, if I remember correctly, MDPI does not have any page limitation (the APC is the same)? The authors may add their response for the 3-user NOMA case in the manuscript.

4.      By the way, in the response report for the previous comment 1, the authors mention different NOMA groups use different time-frequency resource blocks to reduce interference from the other NOMA groups. I suppose some procedures like guard time may be required to guarantee the interference can be fully mitigated? I suggest the authors also include this for their results of spectral efficiency.

Reviewer 3 Report

no more comments.